# Poly(butylene adipate-co-terephthalate)/Polylactic Acid/Tetrapod-Zinc Oxide Whisker Composite Films with Antibacterial Properties

**DOI:** 10.3390/polym16081039

**Published:** 2024-04-10

**Authors:** Zhibo Zhao, Rajkamal Balu, Sheeana Gangadoo, Naba Kumar Duta, Namita Roy Choudhury

**Affiliations:** 1Chemical and Environmental Engineering, School of Engineering, RMIT University, Melbourne, VIC 3000, Australia; s3679391@student.rmit.edu.au (Z.Z.); rajkamal.balu@rmit.edu.au (R.B.); sheeana.gangadoo@rmit.edu.au (S.G.); 2ARC Industrial Transformation Research Hub for Transformation of Reclaimed Waste into Engineered Materials and Solutions for a Circular Economy (TREMS), RMIT University, Melbourne, VIC 3000, Australia

**Keywords:** poly(butylene adipate-co-terephthalate), polylactic acid, tetrapod-zinc oxide, melt-extrusion, blow molding, composite film, antibacterial, packaging material

## Abstract

Biodegradable composite films comprising of poly(butylene adipate-co-terephthalate) (PBAT), polylactic acid (PLA), and tetrapod-zinc oxide (T-ZnO) whisker were prepared by a melt-extrusion and blow molding process. The effect of the incorporation of the T-ZnO whisker (1 to 7 wt.%) in the PBAT/PLA blend film was studied systematically. The composite films with an optimal T-ZnO whisker concentration of 3 wt.% exhibited the highest mechanical (tensile strength ~32 MPa), rheological (complex viscosity~1200 Pa.s at 1 rad/s angular frequency), and gas barrier (oxygen permeability~20 cc/m^2^·day) properties, whereas the composite films with 7 wt.% T-ZnO whiskers exhibited the highest antibacterial properties. The developed composite films can find potential application as antibacterial food packaging materials.

## 1. Introduction

In recent decades, petroleum polymer materials have gained a wide range of applications including plastic film packaging, disposable materials, and agricultural mulching films [1,2]. However, disposable food packaging films and discarded agricultural films made from these polymers present significant negative impacts to the environment. Environmental problems, such as “White pollution” [3], and “Microplastics” [4] caused by the non-biodegradable material have urged researchers to find biodegradable alternatives. Some of the biodegradable polymer materials, such as polylactic acid (PLA) [5], poly(butylene adipate-*co*-terephthalate) (PBAT) [6], and polymethyl ethylene carbonate [7], which can be produced by renewable sources, have become a substitute for non-biodegradable polymer ingredients.

As a commonly used biodegradable polymer, PLA can be prepared from sugar and corn and can be used for biodegradable packaging and medical application. However, PLA has some drawbacks, such as low mechanical strength and thermal stability which could limit its properties [5]. One of the most effective methods to enhance the properties of the PLA is to blend with other flexible polymer materials such as polycaprolactone [8], poly(butylene succinate-co-terephthalate) [9], and PBAT [10]. PBAT composed of butanediol, adipic acid, and terephthalic acid is an ideal candidate to blend with PLA and enhance its impact strength and elongation properties due to its flexibility [10]. However, PLA and PBAT are not compatible, which could be addressed by introducing a compatibilizer or chain extender. The most widely used commercial chain extender is the Joncryl ADR family of copolymers composed of styrene, glycidyl methacrylate, and butyl acrylate. The ADR chain extender is expected to enhance the properties of the blends in film blow molding and the elongation at break [11]. Although the PBAT/PLA blend film has optimized mechanical and biodegradable properties, the high cost and poor barrier performance of PBAT/PLA blends compared to commonly used non-degradable plastics constrain its development and application [12].

The addition of fillers into polymer blends is one of the efficient methods to improve the barrier properties. Fillers like calcium carbonate and nanocrystal cellulose (NCC) are reported to enhance the properties of the PBAT/PLA polymer blend [13]. When filled with nanocrystal cellulose-silver nanohybrids (NCC-Ag), the PBAT/PLA/NCC-Ag composite blend also exhibits antibacterial properties which could be used as potential antibacterial packaging material [14]. ZnO-based nanocomposites of different morphologies (spherical, hexagonal, nanofibers, and core–shell) have been extensively reported in the literature for their enhanced photocatalytic (degradation of methylene blue dye) and antibacterial activity (against *E. coli* and *S. aureus*) [15,16,17]. The tetra-needle-like or tetrapod ZnO (T-ZnO) whisker was discovered in the 1940s and the method of preparation of T-ZnO was first reported in 1990 [18]. Compared with other fillers, T-ZnO whisker can significantly enhance the mechanical, electrical, antibacterial, and UV resistance properties of polymers [19]. Shi et al. [20] reported that the silane coupling agent modified T-ZnO whisker could enhance impact strength of polyamide 6-based composites. Wang et al. [21] fabricated a high-density polyethylene (HDPE)/T-ZnO whisker blend and indicated that 5 wt.% of T-ZnO whisker could enhance the tensile strength of the blend because of the strong interfacial interaction of T-ZnO whisker. While it shows improved performance with non-degradable polymers, T-ZnO whisker-modified PBAT/PLA blend film has not been investigated, which could become one of the replacements of the HDPE film. Therefore, this research aims to focus on how the T-ZnO whisker impacts the processing and performance of the PBAT/PLA film with an optimized ratio of the whisker in the blend film with balanced properties.

## 2. Materials and Methods

### 2.1. Materials

PBAT (KHB21AP11) with a Melt Flow Index (MFI) of 3–5 g/10 min was procured from Kanghui New Material Technology Co., Ltd. (Yingkou, China). PLA (4032D) was procured from Nature Works (Blair, NE, USA). T-ZnO whisker was procured from WOWMATERIALS (Changzhou, China). 3-amino propyltriethoxysilane (KH550) was procured from Aladdin (Shanghai, China). Joncryl ADR-4380 chain extender was procured from BASF. Co., Ltd. (Ludwigshafen, Germany). Film blowing agent (Erucic acid amide) was procured from Jiangsu Runfeng Synthetic Technology Co., Ltd. (Jiangsu, China).

### 2.2. Surface Modification and Morphology of T-ZnO Whiskers

#### 2.2.1. Surface Modification of T-ZnO Whisker

Ethanol was used to dissolve 5 wt.% of the silane coupling agent, and the unreacted silane coupling agent was pre-hydrolyzed at room temperature for about 30 min. The pH of the mixture was adjusted to 4 by adding hydrochloric acid. After pH adjustment, the T-ZnO whisker and ethanol-coupling agent mixed solution was reacted under magnetic stirring in a two-necked round bottom flask fitted with a condenser. The reaction condition was 70 °C for 4 h. After the reaction, the mixture was vacuum-filtered and washed to remove the remaining ethanol. Finally, modified T-ZnO whisker powder was vacuum-dried at 80 °C for 12 h.

#### 2.2.2. Morphology of Unmodified and Modified T-ZnO Whiskers

The morphology of unmodified and modified T-ZnO whiskers was observed using a Gemini 300 scanning electron microscope, SEM (Carl Zeiss AG, Oberkochen, Germany). The samples were sputter-coated with gold (5 nm thick) for SEM analysis.

### 2.3. Preparation of PBAT/PLA/T-ZnO Whisker Composite Blends and Film Samples

#### 2.3.1. Preparation of Composite Blends

Before blending, PBAT and PLA were oven dried overnight. The modified T-ZnO whisker, PLA, PBAT and, ADR chain extender mixture was premixed for 15 min in a Shr-50A Paddle Type Powder Hot Mixer (Zhangjiagang Beierman Machinery Co., Ltd., Zhangjiagang, China) and extruded using a SHJ-20 twin-screw extruder (Nanjing GIANT Machinery Co., Ltd., Nanjing, China). The temperature in the extruder was set from 180 to 190 °C with a gradual increment in temperature from the feed zone to the die zone. After extruding, the melted blends were water-cooled and cut into pellets. The sample name and the composition of the fabricated blend and composites are summarized in Table 1. The T-ZnO whisker content varied from 1 to 7 parts per hundred (phr) parts of polymer in the composite films.

#### 2.3.2. Preparation of Composite Films

PBAT/PLA/T-ZnO whisker sample films were produced using a laboratory small mini film blowing machine (Zhangjiagang Lianjiang Machinery Co., Ltd., Zhangjiagang, China). The temperature of molding was set at 190 °C. Sample films were produced at a low speed of 49 rpm including the speed of the screw traction and reel.

### 2.4. Characterization of PBAT/PLA/T-ZnO Whisker Composite Films

#### 2.4.1. Optical Properties

The optical properties including haze and light transmittance were tested using a haze-gard type 2 Transparency Transmission Haze Meter (Toyo Seiki Seisaku-sho, Ltd., Tokyo, Japan). About 5 samples were tested for each composition and the average number with a standard deviation was taken for the haze and light transmittance value.

#### 2.4.2. Morphology and Elemental Composition

The surface and cross-sectional morphology of the composite films were analyzed using a QUANTA 200 FEG SEM (FEI, Hillsboro, OR, USA) equipped with an energy-dispersive X-ray spectrometer (EDS). The accelerated voltage of the SEM was set at about 20 kV. Before SEM analysis, the films were cryo-fractured and sputter-coated with a 5 nm thin gold layer. The EDS spectra were recorded to analyse the elemental composition of the fabricated composite samples.

#### 2.4.3. Functional Groups

Fourier-transform infrared spectroscopy (FTIR) was used to analyze the functional groups of blend and composite films. The FTIR spectra were recorded in the wavenumber range of 600–4000 cm^−1^ using a Spectrum 100 FTIR (Perkin Elmer, Shelton, WA, USA) equipped with an Attenuated Total Reflectance (ATR) accessory.

#### 2.4.4. Mechanical Properties

The tensile test of fabricated blend and composite films was conducted according to the ISO 527 test standard using a TM105D universal testing machine (Shenzhen Wance Testing Machine Co., Ltd., Shenzhen, China) equipped with a 100 N load cell. The test was performed at a cross-head speed of 100 mm/min. Before the experiment, the samples were cut into dumbbell shapes with a dimension of 115 mm × 25 mm × 6 mm in a film-cutting machine, and the thickness of the sample dumbbell narrow neck was measured and recorded using a micrometer caliper (Yueqing Hemu Instrument Co., Ltd., Yueqing, China). About 5 samples were tested for each composition and the average value with standard deviation was reported for the tensile strength and elongation at the break of the sample.

#### 2.4.5. Thermal Properties

The thermal degradation profile of the fabricated samples was measured by thermogravimetric analysis (TGA) using a Q500 thermal analyzer (TA Instrument, New Castle, DE, USA). The sample weight used for the test was about 5–6 mg, and the test was conducted under nitrogen atmosphere (flow rate 25 mL/min). The temperature range of the test was from room temperature to 600 °C at a heating rate of 20 °C/min.

The change in physical properties of the sample as a function of temperature and time was evaluated by differential scanning calorimetry (DSC) using a Discovery DSC 250 machine (TA Instrument, New Castle, DE, USA). The test was conducted under nitrogen atmosphere (flow rate 25 mL/min) with samples typically loaded in Tzero aluminum pans. The first heating scan was performed from −80 °C to 200 °C to eliminate the processing history, followed by the cooling scan performed from 200 °C to –80 °C to record crystallization and glass transition event, and finally the second heating scan was performed from −80 °C to 350 °C to record glass transition, melting, and degradation events. A scan rate of 10 °C/min was used for all the three steps. The glass transition temperature (T_g_), melting temperature (T_m_), enthalpy of melting (ΔH_m_), crystallization temperature (T_c_), and enthalpy of cold crystallization (ΔH_cc_) were calculated using the TRIOS Software (TA Instrument, New Castle, DE, USA).

#### 2.4.6. Crystallinity and Intrinsic Structure

The crystallinity of PLA, PBAT and the blend was calculated using Equations (1)–(3), as given below [22]:(1)$$\chi_{C}\left(PLA\right)=\frac{𝛥H_{m}-{𝛥H}_{cc}}{W_{PLA}\cdot 𝛥H_{m-PLA}^{O}}\times 100\%$$
(2)$$\chi_{C}\left(PBAT\right)=\frac{𝛥H_{m}}{W_{PLA}\cdot 𝛥H_{m-PBAT}^{O}}\times 100\%$$
(3)$${\;\chi }_{C}\left(Blends\right)={W_{PLA}\;\cdot \;\chi }_{C}\left(PLA\right)+{W_{PBAT}\;\cdot \;\chi }_{C}\left(PBAT\right)$$
where, $\Delta H_{m}$ and ${\Delta H}_{cc}$ are the enthalpies of melting and cold crystallization, W  is the weight ratio of each polymer, and $\Delta H_{m-PLA}^{O}$ is the theoretical melting enthalpy of the polymer when it is fully (100%) crystalline (93 J/g for PLA and 114 J/g for PBAT) [23].

The X-ray Diffraction (XRD) experiment was performed using a D4 Endeavor XRD instrument (Bruker, Billerica, MA, USA). Film samples were cut into pieces of size 2.5 × 2.5 cm^2^ for analysis. The test diffraction angle (2θ) ranged from 5 to 40°, and the scanning speed was 10°/min.

The intrinsic structure of the fabricated films was investigated by the small-angle X-ray scattering (SAXS) technique using a Xeuss 3.0 SAXS instrument (Xenocs, Grenoble, France) at the Australian Centre for Neutron Scattering (Sydney, Australia), operated with an X-ray wavelength of 0.154 nm. The SAXS data was collected in the scattering vector (Q) range of 0.008–0.5 Å^−1^, where $Q=\left(4\pi sin\theta \right)/n\lambda$; θ is the angle of scattering, and *λ* is the wavelength of X-ray. The exposure time for the X-ray was about 60 min for each sample. The long spacing (d_ac_)—defined as the sum of the crystal layer thickness together with one interlamellar amorphous layer, the thickness of the crystal (d_c_), and amorphous (d_a_) layers were calculated using Equations (4)–(6), as given below [23]:(4)$$d_{ac}=\frac{2\pi }{q_{max}}\;$$
(5)$$d_{c}=d_{ac}\;\cdot \;X_{c}\;$$
(6)$$d_{a}=d_{ac}-d_{c}\;$$

#### 2.4.7. Rheological Behavior

The viscoelastic properties of the samples were measured using a Discovery HR3 rheometer (TA instrument, New Castle, DE, USA). The sample was tested at 190 °C, and the storage modulus (G′), loss modulus (G″), and complex viscosity (η*) were recorded as a function of frequency (0.1–100 rad/s). The strain for the test was set to 1% to ensure the samples were tested in the linear viscoelastic region.

#### 2.4.8. Film Barrier Properties

The water vapor permeability (WVP) of fabricated films were measured using a GV-33-E water vapor transmittance tester (Lab-stone, Guangzhou, China). The film samples were cut into a circular shape (diameter of 74 mm) and loaded in the equipment for testing. The test temperature was kept at 38 °C for 3 days.

The oxygen transmittance (OT) properties of the fabricated films were measured using an oxygen permeation testing analyzer (Mocon, Minneapolis, MN, USA). The film samples were cut into a circular shape (diameter of 5 cm) and loaded in the equipment for testing. The measurement values were based on the average of the three replicate specimens with standard deviation.

#### 2.4.9. Antibacterial Properties

The antibacterial activity of the fabricated films was evaluated against Gram-negative bacteria, *Escherichia coli.* A microbial solution of *E. coli* was prepared in Luria–Bertani (LB) broth, and the optical density was measured using a UV-vis spectrometer until it reached ~0.1 at 600 nm (OD_600_). The samples were incubated (in triplicates) with the microbial solution for 24 h at 37 °C, along with positive controls (*E. coli* only). After 24 h incubation, samples were gently washed with sterile phosphate-buffered saline (PBS), stained with 10–20 µL of LIVE/DEAD BacLight^TM^ Viability Kit (including SYTO 9 and propidium iodide) (Molecular Probes ^TM^, Invitrogen, Grand Island, NY, USA) and incubated for 10 min in the dark at room temperature, according to the manufacturer’s protocol. Samples were washed with PBS twice and visualized using a ZEISS LSM 880 Airyscan upright microscope (Zeiss, Oberkochen, Germany). A minimum of two images were taken per replicate and the proportion of live to dead cells was analyzed using ImageJ software 1.54h. Antimicrobial efficacy (%) was calculated using Equation (7), and student’s *t*-test was used to assess the significance.
(7)$$Antimicrobial\;efficacy\;\left(\%\right)=\frac{Dead\;cell}{Total\;cell}\times 100$$

## 3. Results and Discussion

### 3.1. Morphology of T-ZnO Whisker

Figure 1 shows the SEM image of the as procured T-ZnO whisker. The T-ZnO whisker exhibited a unique tetrapod morphology, where the angle between each needle was observed around 109°. Two size fractions of about 5–10 μm (predominant), and 20–40 μm were measured for the T-ZnO whisker, as shown in Figure 1a,b, respectively.

### 3.2. Surface Chemistry and Morphology of Modified T-ZnO Whisker

The direct addition of surface unmodified T-ZnO whisker into polymer matrix has been previously reported to exhibit poor interfacial interaction between the T-ZnO whisker and the polymer matrix resulting in agglomeration, the formation of micro-cracks and subsequently poor mechanical properties [24]. Surface modification by coupling agents (organo-functional materials) is an efficient way to improve the interfacial interaction between inorganic filler and polymer material [25]. Therefore, to improve the interfacial interaction between the T-ZnO whisker and PBAT/PLA blend, the surface of the T-ZnO whisker was modified using a silane coupling agent (KH550). Figure 2a,b illustrates the schematics of hydrolysis reaction of the coupling agent.

The hydrolysis reaction of KH550 silane coupling agent leads to the formation of silanol groups (Figure 2a), which react with the hydroxyl groups on the surface of T-ZnO whiskers to form siloxane link (Figure 2b) and change the surface of T-ZnO whisker from hydrophilic to hydrophobic. During composite mixing, the –NH_2_ group on the surface modified T-ZnO whisker can interact with the PBAT/PLA blend, resulting in good dispersion and interfacial interaction. Figure 2c shows the comparison of pictures between the unmodified T-ZnO whisker powder and the modified T-ZnO whisker powder. The surface morphology of the silane-treated T-ZnO whisker is shown in Figure 3, where compared to the sharp needle-like structures in the unmodified T-ZnO whisker, the silane-treated T-ZnO whisker showed relatively blunt edges, with preferential absorption of the silane coupling agent on the surface of the whisker [26].

### 3.3. Appearance and Optical Properties of Films

Figure 4a shows the T-ZnO whisker filled PBAT/PLA blend sample (3-Zn) successfully blown into film after melt processing. Figure 4b compares the transparency of fabricated films. The transparency of the film could be evaluated by the haze and the light transmittance as well. The haze of a film is defined as the part of transmitted light deviated from the direct transmitted beam by more than 2.5° [27].

Figure 5 shows the haze and light transmittance measured for the neat PBAT/PLA film and T-ZnO whisker filled composite films. The haze and light transmittance of neat PBAT/PLA are about 91.9% and 81.5%, respectively. With the addition of the T-ZnO whisker, the composite films showed a decreasing trend of haze. The haze of 1-Zn, 3-Zn, and 7-Zn are about 91.2%, 78.6% and 73.3%, respectively. This is due to the change in crystallinity of the film with addition of the ZnO whisker. The light transmittance of the neat PBAT/PLA film is about 81%. The addition of the T-ZnO whisker did not show a clear trend in light transmittance. The light transmittance of 1-Zn is about 84.8%, 3-Zn is about 81.6% and the 7-Zn is about 83.2%, where the overall effect is not significant from that of the pure blend. This could be related to the particle size and degree of dispersion of the T-ZnO whisker.

### 3.4. Surface and Cross-Sectional Morphology of Films

The surface and cross-sectional morphology of the PBAT/PLA/T-ZnO whisker composite films as a function of T-ZnO concentration were evaluated using SEM, as shown in Figure 6 (high magnification images). The pure PBAT/PLA blend film (0-Zn) exhibits a rough surface morphology with elongated structures aligned towards the machine direction (MD) due to polymer processing induced plastic deformation. However, with a systematic increase in T-ZnO whisker content, the surface morphology of the composite films became relatively smooth. This can be attributed to the T-ZnO whisker induced reduction in the plastic deformation (stress relaxation) of the polymer blend matrix, where the stress applied to the polymer matrix during thermomechanical processing would transfer into one of the T-ZnO whiskers in the surrounding region and be distributed to other pods across the three-dimensional space [21]. The level of the T-ZnO whisker used in this study did not affect the homogeneity of the films.

Figure 7 shows the low magnification SEM image of the 7-Zn composite film sample which clearly shows uniform distribution of the T-ZnO whisker (white spots) supporting good interfacial interaction between the T-ZnO whisker and PBAT/PLA polymer blend matrix. Furthermore, the EDS results confirm the particles seen in the SEM image to be ZnO, with area EDS spectra showing Zn and O elemental peaks, and spot EDS spectra showing a relatively higher intensity of Zn elemental peak.

### 3.5. Functional Groups of Fabricated Films

FTIR analysis was performed to study the functional groups of the fabricated films. Figure 8 shows the FTIR spectra of the neat PBAT/PLA blend film compared to PBAT/PLA/T-ZnO whisker composite films. The FTIR bands observed around 3000 cm^−1^, 1750 cm^−1^, 1270 cm^−1^, 1089 cm^−1^ and 726 cm^−1^ for the 0-Zn sample (neat PBAT/PLA blend) correspond to the C–C stretching, C=O stretching, symmetrical or unsymmetrical deformation of the C–C, absorption band of C–O, and vibration band of −CH_2_ group, respectively [28,29]. Compared with the 0-Zn sample, no significant change in the position and intensity of peaks was observed for the composite films. A similar observation was also made by Kim et al. [30] for PLA/ZnO composites.

### 3.6. Mechanical Properties of Fabricated Films

The tensile tests of the fabricated films were conducted on both the transverse direction (TD) and machine direction (MD) of the blow molded samples to find out the anisotropy in the mechanical properties. The obtained mechanical properties, such as tensile strength and elongation at break are presented in Figure 9. From Figure 9a, it can be observed that the tensile strength values of the fabricated films in MD is higher than that in the TD. In general, the strength of polymer film in MD is higher than TD because of the effect of longitudinal stretching and the oriented spherulites of polymers in the blow molding process [13]. For the neat PBAT/PLA, the tensile strength is measured around 28 MPa and 13 MPa in the MD and TD, respectively. The addition of the T-ZnO whisker to the PBAT/PLA blend showed an increasing trend in the tensile strength of the composite films, which can be attributed to uniform distribution of the T-ZnO whisker in the PBAT/PLA blend matrix [19], and the ability of the T-ZnO whisker to withstand stress and transfer to the tetra pod needles when the force is applied to one dimension of the needle [21]. However, when the T-ZnO whisker concentration was increased to 7 wt.% (7-Zn sample), the tensile strength of the composite film decreased, which may be due to T-ZnO whisker agglomeration or damage during processing (as shown in Figure 7 SEM image). In the 7-Zn sample, it is likely that the van der Waals force would bundle the T-ZnO whisker together [24]. As a result, there is a certain correlation between the zinc oxide content and the tensile properties of the material, and the addition of a small amount of T-ZnO whisker is seen to enhance the tensile strength of the PBAT/PLA film better than that at a higher concentration. Among all the tested samples, 3-Zn shows the optimized (highest) tensile strength, which is about 32 MPa. This is equivalent to the tensile strength of polypropylene (~35 MPa) and linear low-density polyethylene (~37 MPa), and the fabricated 3-Zn film could be used as a potential food packaging film or mulching film [31].

On the other hand, the elongation at break in TD was observed to be higher than that in the MD (Figure 9b). With an increase in the T-ZnO whisker concentration, the elongation at break was observed to increase, where 7-Zn exhibited the highest elongation at a break of 550%. The increasing trend of elongation at break with the increase in the T-ZnO whisker is attributed to the orientation of the amorphous phase formed during the blow molding process [13].

### 3.7. Thermal Properties of Fabricated Films

Thermal stability determines the processing temperature and the processability of the polymer and its blends with temperature. TGA is one of the methods to evaluate the thermal stability of the polymer material. Figure 10 shows the TGA and DTG (first derivative) curves of the fabricated film samples, and the parameters obtained are summarized in Table 2. It is shown that the decomposition of neat PBAT/PLA shows two steps around 359 °C and 409 °C, which correspond to the decompositions of PLA and PBAT, respectively [32]. With the increase in ZnO content, the degradation onset temperature of composite films decreased. This can be attributed to the selective depolymerization of PLA catalyzed by ZnO, where the first DTG peak was observed to shift significantly to a lower temperature with the increase in intensity, whereas the second peak shifted only slightly with the decrease in intensity [30]. With the addition of 7 wt.% T-ZnO, the DTG peak for PLA in the composite film shifted from 359 °C to 315 °C. Conversely, the amount of residue at 600 °C increased with the increase in ZnO content as it is inert to pyrolysis.

Figure 11 shows the DSC curves of the fabricated film samples at both heating and cooling cycles, and the measured parameters related to melting and crystallization events are summarized in Table 3. For the pure PBAT/PLA blend film (0-Zn), the change in heat flow slope observed around −32 °C and 60 °C are attributed to the glass temperatures (*T_g_*) of PBAT and PLA. With the addition of the T-ZnO whisker, the Tg values of the blends did not change considerably, indicating their immiscibility. The endothermic peak observed around 130 °C and 168 °C are attributed to the melting temperatures (*T_m_*) of PBAT and PLA, respectively. The addition of the T-ZnO whisker did not change the melting temperature of the PBAT and PLA, and the crystallization temperature of PBAT. However, the PLA crystallization peak shifted to a lower temperature in the presence of T-ZnO whisker, which supports interaction of PLA and T-ZnO whisker during the fabrication of composite films.

### 3.8. Crystallinity and Intrinsic Structure of Fabricated Films

The crystallinity of the PBAT and PLA in the blend film is impacted largely by the addition of the T-ZnO whisker. It is shown that a 1–3 wt.% T-ZnO whisker enhances the crystallinity of PBAT and PLA, whereas crystallinity decreases with 7 wt.% T-ZnO whisker concentration. Yu et al. [33] have reported that ZnO inhibits the motion of the PLA chain showing a nucleating effect. The crystallinity (χ_c_) values calculated from the DSC curve melting enthalpy of PBAT and PLA are presented in Table 3, which shows an increase in the crystallinity of PLA with an increase in T-ZnO content, whereas the crystallinity of PBAT both increased and decreased. The 3-Zn sample exhibited the highest overall crystallinity of 12.36%. Figure 12 shows the XRD curves of the neat T-ZnO whisker and all the fabricated composite films. For the T-ZnO whisker, strong diffraction peaks were observed around the 2θ values of 31.9°, 34.4°, and 36.3°, which correspond to the (100), (002), and (101) planes, respectively, of the hexagonal wurtzite-structured ZnO (Joint Committee for Powder Diffraction Standards: 36-1451) [34]. For neat PBAT/PLA blend, diffraction peaks were observed around the 2θ values of 17.7°, 20.7°, 23.3°, and 24.9°, which can be attributed PBAT crystalline peaks [13], whereas the peak observed around 28° corresponds to β-crystalline PLA [35]. For composite films, with an increase in T-ZnO whisker content, the intensity of the peaks corresponding to ZnO around 31.9°, 34.4°, and 36.3° increased systematically, whereas the intensity of the peak at 28° decreases, which suggests intermolecular interaction between PLA and surface modified T-ZnO whisker.

The physicochemical properties of semi-crystalline polymer blends, such as PBAT/PLA largely depend on their crystalline lamellar structure. Small-angle X-ray scattering (SAXS) was used to investigate the lamellar structure of the fabricated films. Figure 13 shows the SAXS intensity profile of the fabricated films in the air, where a broad Lorentzian type of peak on top of a power law decay was observed. The intensity maximum peak value (Q_max_) was obtained by a shape-independent broad peak model fit to the SAXS data using the SasView program. The Q_max_ of PBAT/PLA blend film was observed to decrease significantly with the introduction of a 1 wt.% T-ZnO whisker, which further decreases with an increase in T-ZnO whisker content. The obtained Q_max_ and other structural parameters, such as the thickness of the crystal (d_c_) and amorphous (d_a_) layers derived using Equations (3)–(5), along with the crystallinity (χ_c_) of PBAT (obtained from DSC data), are presented in Table 4.

The incorporation of 1 wt.% T-ZnO whisker into PBAT/PLA polymer matrix showed a significant reduction in Q_max_ value, whereas a further increase in T-ZnO whisker content showed only a small but systematic reduction in Q_max_ value. Compared to PBAT, PLA has very weak SAXS intensity due to the low difference in the density of electrons in the crystal and amorphous phase [23]. Therefore, the crystallinity of PBAT estimated from the DSC data was used to estimate the d_c_ and d_a_ values using Equations (4)–(6). The observed trend in the d_c_ value is in general agreement with the measured DSC crystallinity trend, where the 3-Zn sample exhibited the largest d_c_ value. The 1-Zn and 7-Zn samples exhibited a slightly decreased d_c_ value or increased d_a_ value, which possibly resides in the interlamellar structure of PLA in the respective composites [23].

### 3.9. Rheological Properties of Fabricated Films

The rheological property reflects the processability and blow molding behavior of the polymer blend and composite materials. A polymer material with a higher storage modulus and complex viscosity could have strong melting strength, which is essential for film blowing. Figure 14a,b shows the storage modulus and complex viscosity of neat PBAT/PLA and PBAT/PLA/T-ZnO whisker composite blends measured as a function of angular frequency at 190 °C. The storage modulus of the neat PBAT/PLA blend increases with increasing frequency from 0.1 to 100 rad/s. With the T-ZnO whisker addition, the storage modulus of composite films increases for 1 and 3 wt.% T-ZnO concentration. However, due to the agglomeration of the high content of the T-ZnO whisker, the storage modulus decreases for the 7-Zn sample in the low frequency. However, the storage modulus in the high frequency of the samples is nearly the same due to the breakage of the molecular chain entanglement of PBAT and PLA by the fast shear effect.

The negative trend of complex viscosity is illustrated in the Figure 14b curve with the increasing frequency indicating that PBAT/PLA is a typical shear-thinning material. This is due to the consumption of some epoxy functions by reacting at the material interface [36]. The PBAT/PLA samples with the addition of the 1 and 3 wt.% T-ZnO whisker show higher complex viscosity than that of the neat polymer material when the frequency is low. This is due to the tetra needle stereoscopic shape of the ZnO whisker, which would easily entangle with the PBAT/PLA matrix. The strong entanglement would enhance the rheological properties of PBAT/PLA. However, the complex viscosity of 7-Zn shows a lower value at a low angular frequency compared to 1-Zn and 3-Zn because a large amount of ZnO whisker would agglomerate in the polymer matrix. The agglomeration of the whisker would reduce the entanglement of the polymer matrix and tend to have negative rheological properties. As a result, a lower amount of the T-ZnO whisker would enhance the rheological properties of the PBAT/PLA blends, where 1-Zn shows the best rheological properties compared with other fabricated samples.

### 3.10. Film Barrier Properties of Fabricated Films

The barrier characteristics of polymers are essential property for food packaging film or mulching film applications as it could retard the permeation (entering or leaving) of small molecules such as oxygen and water. This could slow down the nutrient loss, increase the shelf life of food, and shorten the moisture duration of agricultural products. Therefore, the addition of the T-ZnO whisker was considered to enhance the barrier properties of the PBAT/PLA blend film. As shown in Figure 15, the oxygen permeability (OP) and the water vapor permeability (WVP) of the neat PBAT/PLA exhibited the highest values (80 cc/m^2^·day and 12.7 × 10^−14^ g·cm/cm^2^·s·Pa, respectively), which decreased with the addition of the T-ZnO whisker. The composite film with a 1 wt.% T-ZnO whisker showed a significant decrease in both OP and WVP compared to neat PBAT/PLA. This is due to the microstructure and uniform morphology of the fabricated film, where the homogeneous polymer network could decrease the diffusion of the oxygen and water vapor and the T-ZnO whisker in the polymer film could interact with PBAT and PLA to reduce the free hydroxyl in the blend film so that the composite film would reduce the water affinity [37]. Increasing the T-ZnO whisker concentration to 3 wt.% increased the permeability but further decreased with the increase in the T-ZnO whisker concentration to 7 wt.%. Due to its hydrophilicity, the ZnO whisker could impact the polymer-filler interactions in the blend which could weaken the cohesion force in the polymer network to increase the WVP of the film [38]. The observed trend in WVP is also in line with SAXS results (long spacing = thickness of the crystal + amorphous layers). Also, Phothisarattana et al. [37] reported that the non-homogeneity and nanovoids in the film raise the diffusion and permeation of the gas molecular which could increase the oxygen permeability of the polymer film. Thus, the use of T-ZnO whisker could increase the barrier properties of the PBAT/PLA blends making them suitable for vegetable and fruit packaging applications.

### 3.11. Antibacterial Properties of Fabricated Films

The antibacterial properties of the fabricated films were investigated against Gram-negative *E. coli* using confocal microscopy, where a quantitative assessment of the % dead cells was evaluated for each film. The Live/Dead staining kit used in the experiment consisted of SYTO 9 and Propidium Iodide (PI). SYTO 9 is a permeable dye capable of penetrating live and dead cells, binding with nucleic acids, and it also fluoresces green. The PI dye is cell impermeable and can only enter cells with damaged membranes, and since it has a stronger affinity to nucleic acids than SYTO 9, it replaces the live dye, binds to nucleic acids, and fluoresces red [39]. Therefore, the green fluorescence shows the live cells, and the red fluorescence represents dead cells. Figure 16 shows representative images of the different film samples taken with the confocal microscopy and Figure 17 demonstrates the antibacterial performance of the films with 57%, 62%, 52%, and 95% of antimicrobial activity expressed with 0 wt.%, 1 wt.%, 3 wt.%, and 7 wt.% ZnO, respectively. The two materials that showed statistically significant differences in antimicrobial efficacy were 3 wt.% and 7 wt.% ZnO containing composite films, with *p* < 0.05 and *p* < 0.001, respectively. It can be concluded that with increasing concentrations of T-ZnO whisker in PBAT/PLA film increases the overall antimicrobial activity of the material.

Previous research on the antimicrobial activity of ZnO has consistently shown its efficacy in killing various types of microorganisms, including Gram-negative bacteria, Gram-positive bacteria, and fungi. Gram-positive *S. aureus* bacteria were reported to be more resistant to ZnO nanoparticles compared to Gram-negative *E. coli* [40]. Therefore, in this work, we tested the antibacterial properties on only Gram-negative bacteria as an initial analysis. ZnO exhibits its antimicrobial activity by three main methods: generation of reactive oxygen species (ROS), physical interaction and shearing of the cell membrane, and release of metal Zn^2+^ ions [41]. ROS may be the primary mechanism of action for Gram-negative bacteria like *E. coli*, as previous research demonstrates that positively charged H_2_O_2_ interacts with the Amide groups of its negatively charged cell membrane, resulting in the breaking down of the membrane and eventually cell death [42].

## 4. Conclusions

In summary, silane coupling agent modified T-ZnO whisker has been successfully incorporated (at various concentrations) into the films of biodegradable PBAT/PLA polymer blend, which remains immiscible. The incorporation of T-ZnO whisker enhanced (by the nucleation effect) the crystallinity of PBAT/PLA films, whereas it affected the optical properties of PBAT/PLA film significantly. An optimal concentration of 3 wt.% T-ZnO whisker was observed to enhance the mechanical and rheological properties of the composite films with strong interfacial interaction between the whisker filler and the polymer blend matrix. On the other hand, the T-ZnO whisker at 7 wt.% concentration exhibited improved barrier properties and strong antibacterial activity of the fabricated composite films. The developed PBAT/PLA/T-ZnO whisker composite films can be used as potential antibacterial packaging material platform.

## Figures and Tables

**Figure 1 polymers-16-01039-f001:**
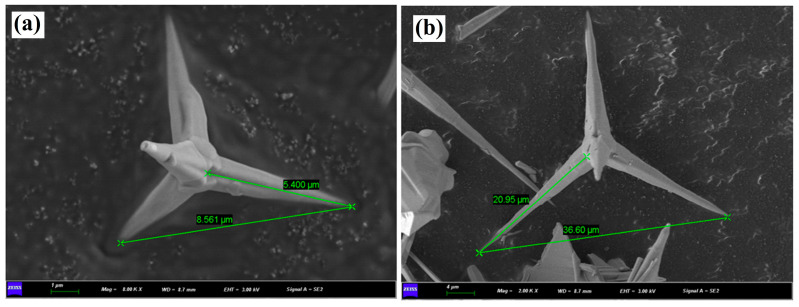
SEM images of tetra-needle or tetrapod like ZnO whisker showing two size fractions: (**a**) 5–10 µm (predominant), and (**b**) 20–40 µm. Scale bar is 1 and 4 µm, respectively.

**Figure 2 polymers-16-01039-f002:**
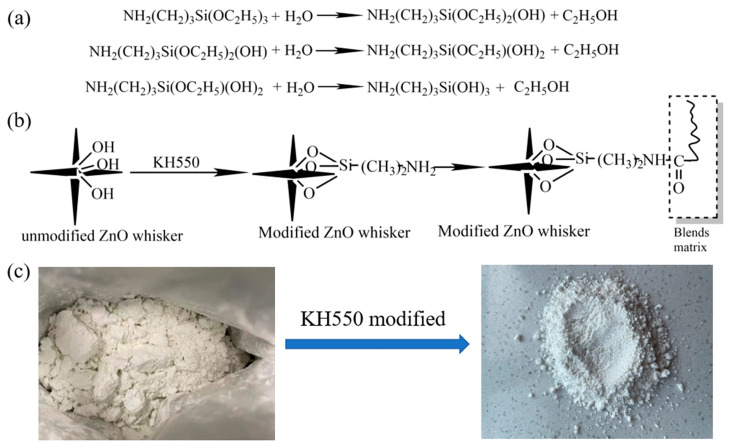
(**a**) Hydrolysis reactions of KH550 silane coupling agent, (**b**) reaction schematic of T-ZnO whisker modified by KH550, and (**c**) pictures of un-modified T-ZnO whisker and modified T-ZnO whisker powders.

**Figure 3 polymers-16-01039-f003:**
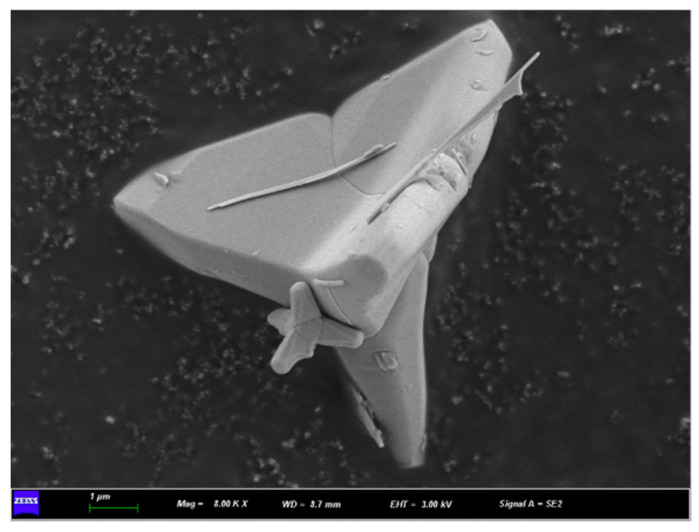
SEM image of surface modified tetra-needle or tetrapod like ZnO whisker. Scale bar is 1 µm.

**Figure 4 polymers-16-01039-f004:**
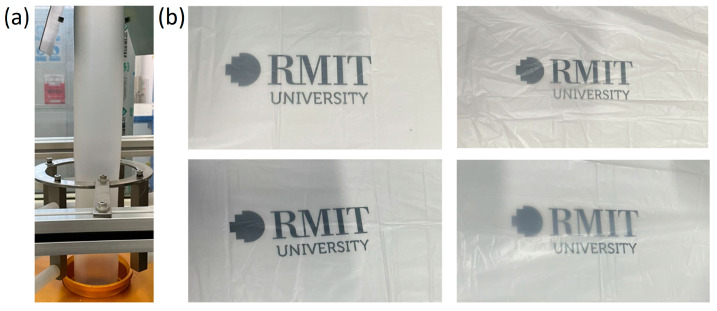
(**a**) Film blowing of PBAT/PLA/T-ZnO whisker composite sample (3-Zn), and (**b**) pictures of the transparency of fabricated film samples (**top left**—0-Zn; **top right**—1-Zn; **bottom left**—3-Zn; **bottom right**—7-Zn).

**Figure 5 polymers-16-01039-f005:**
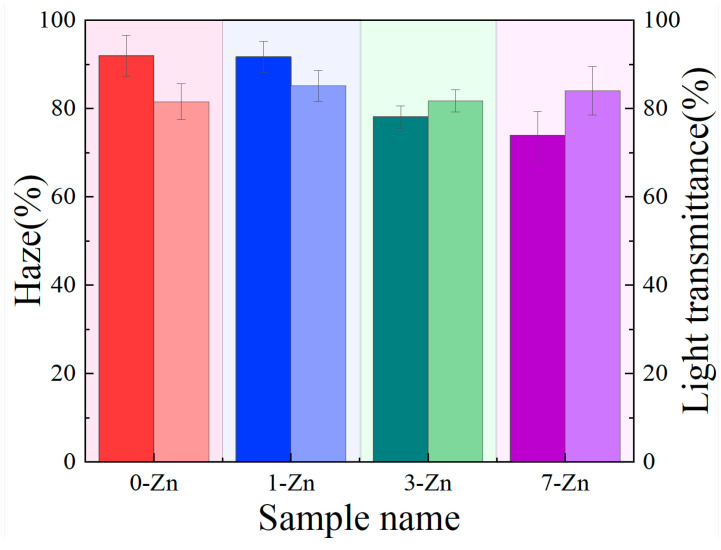
Bar chart of haze (dark color) and light transmittance (light color) measured for fabricated film samples.

**Figure 6 polymers-16-01039-f006:**
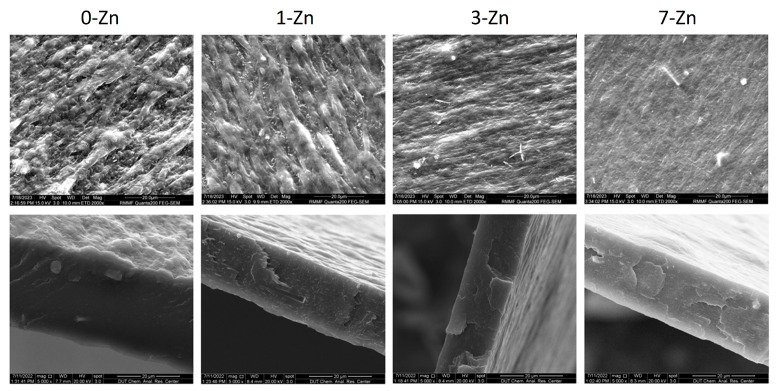
SEM images of the surface (**top**) and cross-sectional (**bottom**) morphology of fabricated film samples. The scale bar is 20 µm.

**Figure 7 polymers-16-01039-f007:**
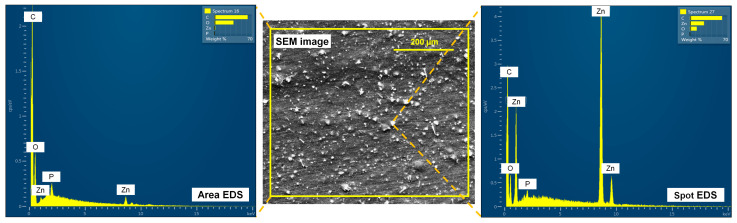
SEM image and corresponding EDS spectra (area and spot analysis) of 7-Zn film sample.

**Figure 8 polymers-16-01039-f008:**
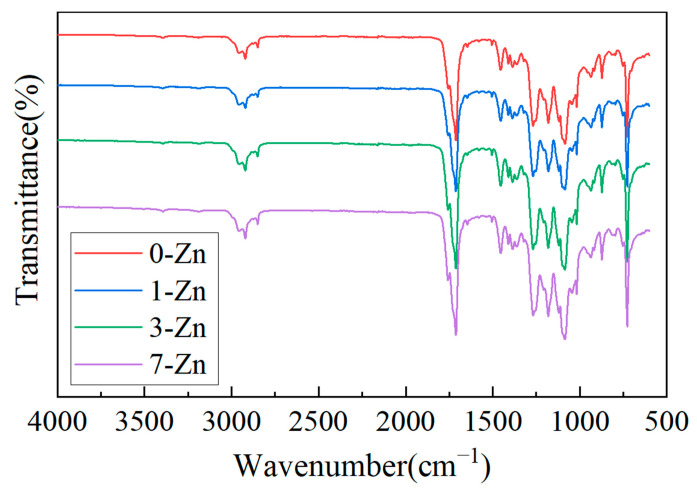
FTIR spectra of fabricated film samples.

**Figure 9 polymers-16-01039-f009:**
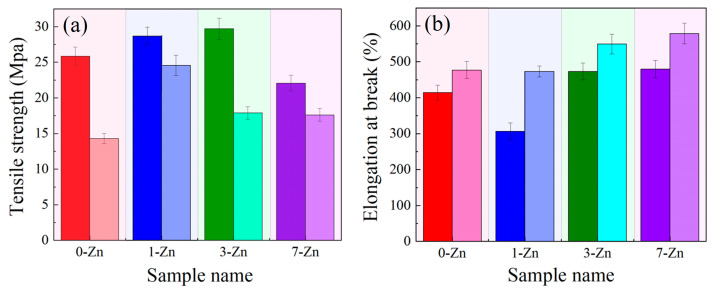
Bar chart of mechanical properties: (**a**) tensile strength, and (**b**) elongation at break (dark bars—MD; light bars—TD) of fabricated film samples.

**Figure 10 polymers-16-01039-f010:**
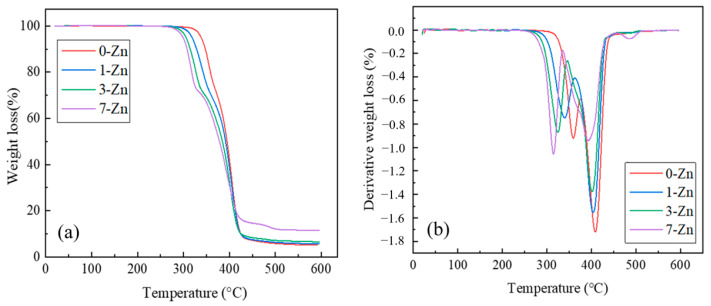
(**a**) TGA, and (**b**) DTG curves of fabricated film samples.

**Figure 11 polymers-16-01039-f011:**
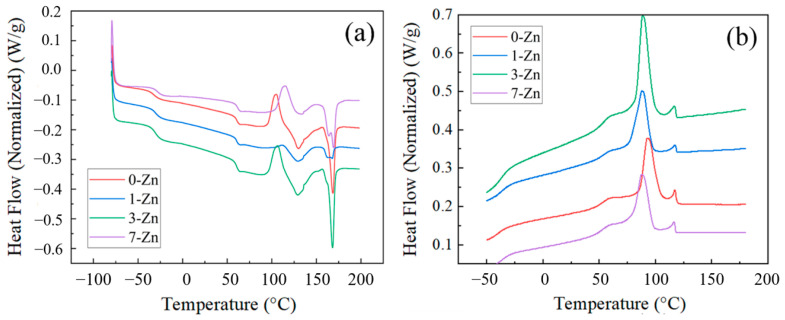
DSC thermograms of fabricated film samples: (**a**) heating, (**b**) cooling cycle.

**Figure 12 polymers-16-01039-f012:**
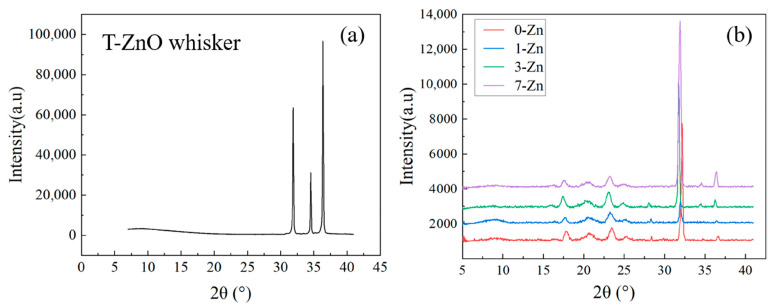
XRD curves of (**a**) T-ZnO whisker, and (**b**) fabricated film samples.

**Figure 13 polymers-16-01039-f013:**
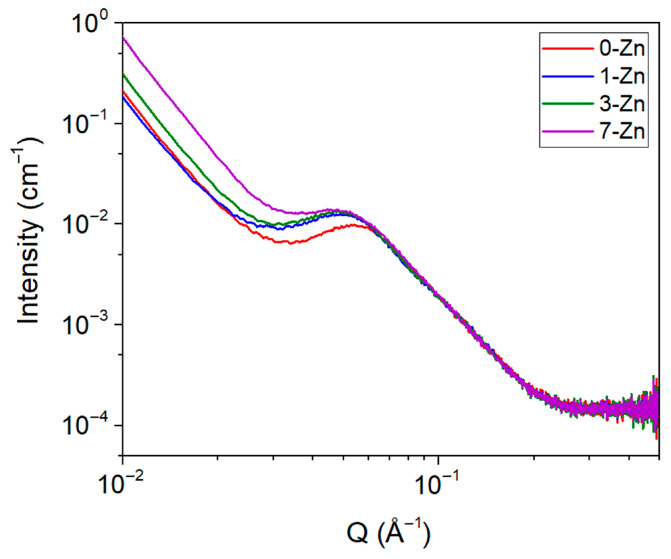
SAXS intensity profile of fabricated film samples.

**Figure 14 polymers-16-01039-f014:**
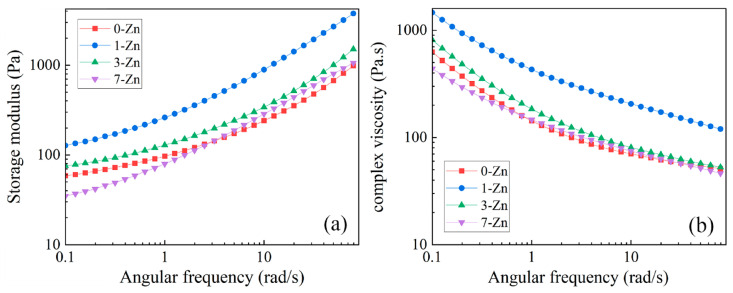
(**a**) Storage modulus, and (**b**) complex viscosity of neat PBAT/PLA (0-Zn) and PBAT/PLA/T-ZnO whisker composite blends measured as a function of angular frequency.

**Figure 15 polymers-16-01039-f015:**
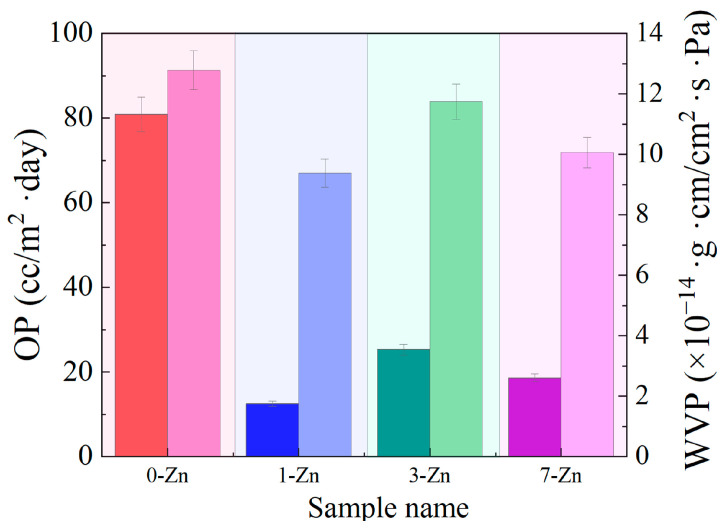
Barrier performance (dark bar—OP; light bar—WVP) of fabricated film samples.

**Figure 16 polymers-16-01039-f016:**
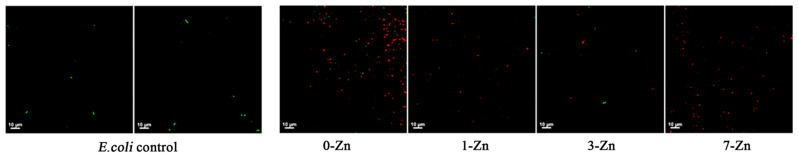
Microscopic images of Live-Dead assay of fabricated film samples. Scale bar is 10 µm.

**Figure 17 polymers-16-01039-f017:**
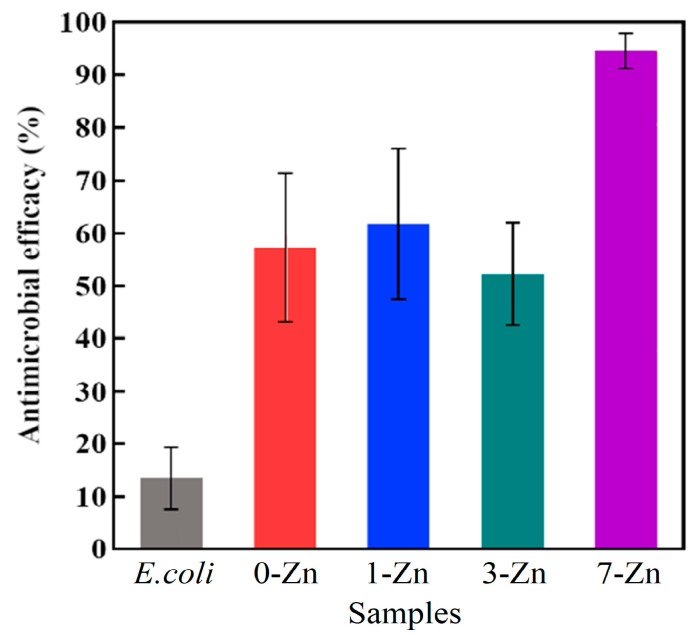
Antimicrobial efficacy of fabricated film samples.

**Table 1 polymers-16-01039-t001:** Composition of fabricated blend and composite samples.

Sample Name	PLA (wt.%)	PBAT (wt.%)	T-ZnO Whisker (phr)	Joncryl ADR-4380 (phr)
0-Zn	30	70	0	0.5
1-Zn	30	70	1	0.5
3-Zn	30	70	3	0.5
7-Zn	30	70	7	0.5

**Table 2 polymers-16-01039-t002:** Summary of TGA results of fabricated film samples.

Sample Name	Degradation Onset Temperature (°C)	Derivative Peak of PLA (°C)	Derivative Peak of PBAT (°C)	Residue at 600 °C (wt.%)
0-Zn	338.6	359.2	408.7	4.8
1-Zn	317.8	340.0	404.2	5.7
3-Zn	305.7	324.8	401.7	7.2
7-Zn	303.7	315.0	393.0	11.1

**Table 3 polymers-16-01039-t003:** Summary of DSC results of fabricated film samples.

Sample Name	T_g_ of PLA (°C)	T_g_ of PBAT (°C)	T_m_ of PLA (°C)	T_m_ of PBAT (°C)	ΔH_m_ of PLA (J/g)	ΔH_cc_ of PLA (J/g)	ΔH_m_ of PBAT (J/g)	χ_c_ of PLA (%)	χ_c_ of PBAT (%)	χ_c_ of Blends (%)
0-Zn	60.28	−32.19	168.52	130.11	8.262	6.955	7.252	4.65	9.09	7.76
1-Zn	60.41	−30.13	167.89	129.19	2.171	0.283	4.924	6.72	6.17	6.34
3-Zn	59.96	−30.62	168.15	129.24	9.085	5.470	9.685	12.87	12.14	12.36
7-Zn	61.33	−28.10	169.51	133.23	8.389	4.634	2.861	13.37	3.59	6.52

**Table 4 polymers-16-01039-t004:** Summary of SAXS results of fabricated film samples.

Sample Name	Q_max_ (Å)	d_ac_ (Å)	χ_c_ (%)	d_c_ (Å)	d_a_ (Å)
0-Zn	0.05416	116.01	7.76	9.00	107.00
1-Zn	0.04853	129.47	6.34	8.20	121.26
3-Zn	0.04845	129.69	12.36	16.03	113.66
7-Zn	0.04838	129.86	6.52	8.46	121.15

## Data Availability

Data are contained within the article.

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
