# Peer review of "Poly(butylene adipate-co-terephthalate)/Polylactic Acid/Tetrapod-Zinc Oxide Whisker Composite Films with Antibacterial Properties"

_polymers, 2024, doi:10.3390/polym16081039_

Round 1

Reviewer 1 Report

Comments and Suggestions for Authors

The manuscript focuses on how the t-ZnOw impacts the processing and performance of the PBAT/PLA film with an optimized ratio of whisker in the PBAT/PLA film with balanced properties. The obtained results sound good. However, there are some points, which should be clarified.

       1. The English should be carefully revised throughout a manuscript.

       2. Do not use abbreviation in title

       3. The reference should be redone as the Journal format

       4. Scheme on page 6 should be added caption and improve the quality.

       5. All error bars are similar. Please check it out

       6. To confirm the formation of PBAT/PLA/ZnO, the SEM-EDS should be provided. 

       7. In Figure 8, with increasing ZnO content, the starting point for temperature decomposition decreases. However, at the last temperature decomposite, the remained content did not make sense with ZnO content. How to explain this concern?

       8. XRD results should be carefully explained. The referee could not find what is the main difference between samples with and without ZnO. Note that, several peaks disappear in ZnO content of 7 as the theta of around 28. The peak at theta of 35 disappeared for ZnO content of 1 however it can be easily found with other ZnO contents.

       9. The related works such as Adv. Nat. Sci: Nanosci. Nanotechnol. 11, 2020, 015009; Adv. Nat. Sci: Nanosci. Nanotechnol. 12, 2021, 045002; Adv. Nat. Sci: Nanosci. Nanotechnol. 14, 2023, 045003 should be referred to introduction.

This manuscript can be considered for publication only when the above-mention questions were especially stressed in the revised manuscript. The referee would like to review a revised version of this paper in the future.

Comments on the Quality of English Language

   The English should be carefully revised throughout a manuscript

Author Response

Thank you for your comments and suggestions on the manuscript. Please find my response attached.

Reviewer 2 Report

Comments and Suggestions for Authors

1. Section 2.3.1. Was the ZnO distributed sufficiently evenly after extrusion? Have any additional manipulations been carried out to mix the composite?

2. Is there a size distribution of ZnO particles? Which ZnO fraction predominates?

3. Page 6. Drawing with a diagram on the page without any signature. Must be signed or removed.

4. Figure 4 may be worth dividing into two separate figures (photos and graph separately). For convenience, it is worth placing a legend on the figure for the histogram.

5. Figures 7, 13. It is also worth adding a legend to the figure for the convenience of the reader.

6. Why is the dependence of WVP on the ZnO content such a dependence. What explains the rise and fall of WVP as content increases?

7. Figure 14 should be divided into two parts (photos and graph separately) and made larger, and also add a legend to the histogram.

8. Why were antibacterial properties tested only on gram-negative bacteria?

Author Response

(The authors gave the same response as above.)

Round 2

Reviewer 1 Report

Comments and Suggestions for Authors

The authors have done their best to improve the quality of this manuscript, which can be published in the present form

Author Response

A file with the response is now attached.
